# [Short Paper] Biomedical Evidence Retrieval with Agentic RAG and Dual Text Encoders

**Dhruv Goyal**[† 1]     **Ema Seibert**     **Ryan Ding**     **Matteo Migliarini**[†]     **Kevin Zhu**[†]

**[†] Algoverse AI Research [1] Indian Institute of Technology Bombay**

## Abstract

We propose an agentic RAG framework for biomedical evidence retrieval that uses iterative query refinement across PubMed and MIMIC-IV clinical notes. Using dual domain-specific encoders and self-critique loops, our system achieves competitive results on PMC-Patients and PubMedQA benchmarks, demonstrating the value of adaptive retrieval for clinical decision support.

## 1 Introduction

Retrieval-Augmented Generation (RAG) has emerged as a leading approach for evidence-based retrieval, combining dense retrieval with generation [Lewis et al., 2020]. In medicine, this paradigm was adapted using domain-specific models like BioBERT to handle specialized terminology [Lee et al., 2020, Gu et al., 2021], yet traditional RAG pipelines are often static, retrieving once without adapting their reasoning.

A more advanced paradigm, Agentic RAG, extends this by embedding autonomous decision-making and iterative reflection into the retrieval loop [Singh et al., 2025]. These systems use agentic control flows, such as corrective feedback or query routing, to achieve more adaptive and reliable reasoning [Yan et al., 2024, Jeong et al., 2024]. To address the need for structured evaluation in this area, this work benchmarks an agentic RAG framework on established biomedical QA datasets [Jin et al., 2019, Tsatsaronis et al., 2015, Pal et al., 2022] and the Patients-PMC benchmark [Zhao et al., 2023] to assess its generalization for clinical cohort discovery.

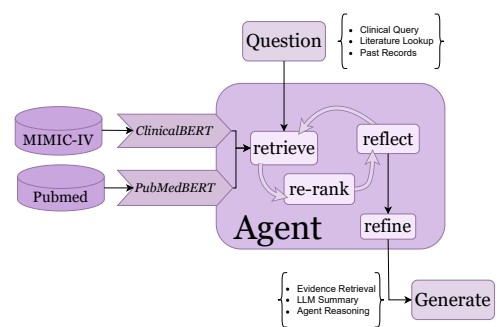

Figure 1: Hybrid biomedical RAG with iterative self-critique. Evidence from PubMed (literature) and MIMIC-IV (clinical notes) is retrieved via domain-specific encoders and re-ranked. An agent cycles between reflect and refine, yielding a final, evidence-grounded response.

## 2 Methodology

Our system employs an agentic RAG framework that iteratively refines search queries and integrates evidence from biomedical literature (PubMed) and clinical notes (MIMIC-IV). The core is a dual-encoder retrieval pipeline orchestrated by an agentic control loop (Figure 1). We encode queries and documents using two specialized models: PubMedBERT for literature and ClinicalBERT for clinical notes, enabling parallel searches Gu et al. [2021], Alsentzer et al. [2019]. Retrieved documents are then merged and refined using a cross-encoder reranker.

Submitted to 39th Conference on Neural Information Processing Systems (NeurIPS 2025). Do not distribute.

Instead of a single pass, an agentic loop assesses evidence quality. If deemed insufficient, the agent triggers a refinement action before re-querying, employing two main strategies: **Pseudo-Relevance Feedback (PRF)**, which refines the query embedding using top-ranked documents, and **Query Decomposition** for complex questions. The loop terminates upon result convergence or after a fixed number of iterations. Finally, a large language model (LLM) synthesizes the refined evidence into a concise, cited answer. Our full code is available at `https://github.com/Dhruv-Git21/Agentic-Biomedical-Retrieval-System`.

# 3 Results

We evaluate our agentic retrieval system on the *PMC-Patients* benchmark—covering Patient-to-Article Retrieval (PAR) and Patient-to-Patient Retrieval (PPR) Zhao et al. [2023]—and the reasoning-free setting of PubMedQA Jin et al. [2019].

As shown in Table 1, our framework achieves competitive results across all tasks. On the PAR task, the system attains high performance. This strong result is expected, as PAR is a known-item retrieval task where high semantic overlap exists between the patient description and the target article. While the model also performs competitively on the more challenging PPR task, the PAR scores highlight the system's strength in precise evidence matching.

On PubMedQA, our framework attains an accuracy of 82.09%, outperforming key baselines such as BioBERT (80.80%). This demonstrates its effectiveness on standard biomedical question-answering benchmarks Table 2.

Table 1: Results for Patient-to-Article Retrieval (PAR) and Patient-to-Patient Retrieval (PPR) on the PMC-Patients dataset. Best results are in **bold**, second best are in *italics*.

| Method | Patient-to-Article (PAR) | | | | Patient-to-Patient (PPR) | | | |
|---|---|---|---|---|---|---|---|---|
| | **MRR@10** | **nDCG@10** | **P@10** | **R@1K** | **MRR@10** | **nDCG@10** | **P@10** | **R@1K** |
| **Agentic (Ours)** | **85.23** | **40.74** | *13.82* | *65.92* | *24.81* | **22.41** | *6.02* | *78.32* |
| SciMult-MHAExpert | *64.44* | *28.62* | **22.12** | **69.09** | **25.35** | *22.39* | **6.65** | **83.78** |
| BM25 | 48.22 | 15.28 | 9.97 | 30.64 | 22.86 | 18.29 | 4.67 | 69.66 |
| Contriever | 15.03 | 4.62 | 3.41 | 16.74 | 10.50 | 8.01 | 2.24 | 52.64 |
| SentBERT | 10.58 | 3.53 | 2.71 | 13.52 | 5.28 | 3.88 | 1.17 | 37.55 |

Table 2: Comparison of reasoning-free baselines on the PubMedQA dataset.

| Model | Acc | F1 |
|---|---|---|
| **Agentic (Ours)** | **82.09** | *62.81* |
| Shallow Features Jin et al. [2019] | 54.44 | 38.63 |
| BiLSTM Jin et al. [2019] | 71.46 | 50.93 |
| ESIM w/ BioELMo Jin et al. [2019] | 74.06 | 58.53 |
| BioBERT Jin et al. [2019] | *80.80* | **63.50** |
| PubMedBERT Gu et al. [2020] | 55.84 | - |
| BioLinkBERT Yasunaga et al. [2022] | 70.20 | - |
| BioLinkBERT-large Yasunaga et al. [2022] | 72.18 | - |
| BioGPT Luo et al. [2022] | 78.20 | - |

# 4 Conclusion

In this work, we demonstrated the effectiveness of an agentic RAG framework for complex biomedical retrieval. Our system achieved competitive performance on the PMC-Patients and PubMedQA benchmarks, highlighting the advantages of agentic strategies over static pipelines. By enhancing retrieval precision and adaptability, these systems represent a promising path toward developing more reliable tools for evidence-based medicine.

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
