# OpenReview forum: "[Short Paper] Biomedical Evidence Retrieval with Agentic RAG and Dual Text Encoders"
_NeurIPS.cc/2025/Workshop_Mexico_City/NORA — NeurIPS 2025 Workshop NORA Poster_

### Official Review · Reviewer_o3Jt · 2025-11-03
**Review on Submission 17**

**Rating:** 7
**Confidence:** 3

**Review:**

Please provide an evaluation of the quality, clarity, originality and significance of this work, including a list of its pros and cons (max 200000 characters). Add formatting using Markdown and formulas using LaTeX. For more information see https://openreview.net/faq

This manuscript is a short paper on the implementation and evaluation of an agentic Retrieval-Augmented Generation (RAG) framework with encoders and self-critique loops, for the purpose of biomedical evidence retrieval. In particular, the PubMedBERT and ClinicalBERT encoders were applied to the PubMed and MIMIC-IV databases respectively and in parallel, for an incoming query. The retrieved text from both sources is then processed within an agentic loop. The raw output from the refinement loop is finally processed by an LLM to give the final answer for human consumption.

- Quality: Implements an agentic loop incorporating input from two large and well-known biomedical databases, with comprehensive comparison against baselines

- Clarity: The manuscript is generally clearly written as a high-level overview of the proposed RAG-and-agent-based retrieval system

- Originality: Agentic loop concept is known but not widely applied to the domain yet; lack of technical details made novelty difficult to judge conclusively

- Significance: Accurate biomedical question answering is a vital subject


Some additional issues might be considered:

1. For the agentic loop, it would be helpful to have details on the Pseudo-Relevance Feedback and Query Decomposition functionalities, as well as the loop iteration limit hyperparameter, and possibility of loop (score) divergence.

2. It could be clarified as to whether the competing methods in Table 1 were reimplemented, or the results obtained from previous publications.

3. The significance of the MRR (mean reciprocal rank), nDCG (normalized discounted cumulative gain), P@10 and R@1K metrics could be analyzed in greater depth, especially against the closest-performing SciMult-MHAExpert. However this might be understandable given the limited page count.

4. Ablation results for only the Pubmed or MIMIC-IV RAG input pipelines would establish the respective advantage of including both sources. Likewise, ablation on agentic loop activation would be relevant given that the GitHub implementation states that the agentic loop is optional.

---

### Official Review · Reviewer_5WLX · 2025-11-03
**Paper introduces an agentic RAG framework for biomedical applications**

**Rating:** 5
**Confidence:** 4

**Review:**

Strengths:
1. High performance on biomendical retrieval and QA tasks using this method
2. Proposed Pseudo-relevance feedback and Query decomposition techniques is a strong implementation of agentic RAG frameworks

Weaknesses:
1. Few details on the proposed techniques, as a result of being a short paper
2. The novelty of the proposed technique is not entirely apparent to me-- query decomposition is well used in agentic RAG or search mechanisms. PRF is also a well-known technique, and it is not clear if anything else is being proposed

---

### Official Review · Reviewer_yvXP · 2025-11-06

**Rating:** 6
**Confidence:** 4

**Review:**

The paper uses dual domain-specific encoders and self-critique loops to build an agentic RAG for biomedical evidence retrieval. It achieves competitive results on benchmarks.

Pros: This is an important research direction on generative retrieval for the biomedical area. The paper did a good exploration in this direction. It builds the due encoder that uses both literature and clinical notes to retrieve the domain specific information. Experiments also show good results on benchmark dataset on PAR and PPR.

Cons: in the agentic loop that assess evidence quality, recommend some search results can also be utilized. Also the efficiency of the proposed framework also needs to be analyzed.  Also wondering, what’s the advantage of the dual encoders? How did the MIMIC-IV and Pubmed interact with each other? Ablation studies on the agent vs directly using LLM as the judge would be interesting.

---

### Official Review · Reviewer_g8Ld · 2025-11-07
**Biomedical Evidence Retrieval with Agentic RAG and Dual Text Encoders**

**Rating:** 7
**Confidence:** 4

**Review:**

The paper proposes an agentic RAG framework for biomedical evidence retrieval that combines iterative query refinement and dual domain encoders (PubMedBERT and ClinicalBERT). It applies loops that automatically improves for adaptive retrieval across PubMed and MIMIC-IV, evaluated on PMC-Patients and PubMedQA benchmarks. Results show consistent improvement over BioBERT and baseline methods.

Strengths:
The problem in adaptive biomedical retrieval is important
Dual-encoder setup is reasonable for mixed-domain data.
The workflow is presented briefly and reproducible.

Weaknesses:
No ablation or sensitivity study to isolate which component contributes to the observed gains.
Reported improvements 82.09% vs. 80.8% are marginal and lack statistical significance, but acceptable
Methodological description omits important implementation details such as embedding dimensions, iteration limits, or reranker brief details..
There is no analysis of computational cost introduced by agentic refinement.

Ethical considerations:
The system deals with sensitive biomedical data and should clearly state its intended use as a support tool, not for diagnosis. Ethical aspects like data privacy (MIMIC-IV compliance), potential bias in pretrained models, and the need for human oversight are important and should be discussed. No direct ethical violations, but the paper lacks reflection on fairness and accountability.

---

### Official Review · Reviewer_vejq · 2025-11-07
**Biomedical Evidence Retrieval with Agentic RAG and Dual Text Encoders**

**Rating:** 6
**Confidence:** 4

**Review:**

This paper presents evaluation results for an Agentic RAG framework on established biomedical QA datasets and the Patients-PMC benchmark. The proposed system uses an agentic loop to assess the quality of the retrieved passages for a query. As part of the loop, the agent triggers a refinement action before re-querying, employing two main strategies: Pseudo-Relevance Feedback (PRF), which refines the query embedding using top-ranked documents, and Query Decomposition for complex questions.

Shortcomings:
Quality/Significance/Impact: The paper is well written but is missing some information that can help decide the quality/impact of this work. It is crucial to understand the response time and cost impact of the agentic loop. More specifically, I would like to know the increase in response time, number of LLM requests, and number of tokens consumed per LLM request. This information gives the reader the full picture on whether the potential improvements of an agent RAG solution over a traditional embedding similarity based or BM is appropriate for their use case.

Novelty/Originality: There are various similar/related work in the area of agentic RAG and more specifically in the biomedical domain. However, I do agree that these related publications might not have the same agent RAG architecture nor evaluate on the same biomedical benchmarks.

---

### Official Review · Reviewer_phV3 · 2025-11-07
**Strong engineering, but need for deeper evaluation**

**Rating:** 6
**Confidence:** 4

**Review:**

The paper proposes an “agentic RAG” pipeline for biomedical retrieval that iteratively refines queries across PubMed and MIMIC-IV, using domain-specific dual encoders (PubMedBERT, ClinicalBERT), a cross-encoder reranker, and a reflect–refine control loop. It reports competitive results on PMC-Patients (PAR, PPR) and PubMedQA.


Strengths

* Clear problem motivation: static one-shot RAG underperforms in clinical settings; iterative retrieval is sensible.
*  Practical system design: dual-encoder per domain + reranker is a reasonable, modular architecture.
*  Cross-domain evidence sources (literature + clinical notes) are valuable for CDS contexts.
*  Competitive PubMedQA accuracy and solid PAR/PPR metrics suggest the approach is effective.

Weaknesses and concerns

*    Clarity/reproducibility:
      -  Details of the agent loop (termination criteria, critique signals, exact PRF formulation, query decomposition algorithm) are underspecified.
       -  How the two encoders’ result sets are merged and normalized (score calibration across domains) isn’t described.
      -  No ablations: impact of PRF vs. decomposition vs. reranking vs. dual-encoder choice is unknown.
*    Evaluation:
      -  “Competitive” claims lack statistical significance tests and variance over seeds.
      -  PAR is near known-item retrieval; high scores there don’t necessarily validate the agentic loop. Stronger emphasis should be on PPR and more challenging QA variants (reasoning-required PubMedQA).
       - Limited baselines: modern retrieval baselines (e.g., ColBERTv2, E5, SPLADE, coCondenser, recent domain LLM retrievers) and agentic/RAG controllers are missing.
       - No efficiency analysis: iteration counts, latency, and compute cost matter for clinical settings.
* Method novelty:
      -  The combination (dual encoders + reranker + PRF + loop) is incremental; conceptual novelty over existing agentic-RAG/controllers is modest. The contribution would be stronger with principled self-critique signals or adaptive stopping policies.
*    Ethics and safety:
      -  No discussion of clinical risk, hallucination control, or provenance fidelity in the generated answers.



My overall recommendation is 6/10 , weak accept. Solid engineering, promising results, and practical relevance to biomedical IR/CDS. However, the novelty is incremental and the paper lacks critical ablations and methodological detail for a stronger endorsement. Strengthen evaluation and clarify the agentic loop to elevate impact.

---

### Official Review · Reviewer_iYAP · 2025-11-07

**Rating:** 6
**Confidence:** 3

**Review:**

This work proposed an agentic RAG framework for biomedical evidence retrieval. It iteratively queries refinement across PubMed and MIMIC-IV clinical notes. The core part of the system is a dual-encoder retrieval pipeline by an agentic control loop. The study evaluated the performance on a set of PMC-Patients and PubMedQA benchmarks, which show a better performance compared to existing baselines.

+ The topic of this work is both relevant and timely, making it of interest to the community.
+ The proposed method should exhibit good performance on a set of commonly used benchmarks.

- The study evaluated the performance of the proposed method against a set of baseline approaches, including shallow features, BiLSTM, BioBERT, BioGPT - for a stronger baseline comparison, maybe a stronger baseline with RAG approach could be tested as well.